# Local-Scale Groundwater Sustainability Assessment Based on the Response to Groundwater Mining (MGSI): A Case Study of Da'an City, Jilin Province, China

**Zhang Fang** [1,2,*] [iD], **Xiaofan Ding** [1,2,3] **and Han Gao** [1,2]

1   Key Laboratory of Groundwater Resources and Environment, Ministry of Education, Jilin University, Changchun 130021, China; dingxf2515@163.com (X.D.); gaohanhan18@163.com (H.G.)
2   Jilin Provincial Key Laboratory of Water Resources and Environment, Jilin University, Changchun 130021, China
3   Shenyang Academy of Environmental Sciences, Shenyang 110004, China
*   Correspondence: azhang9456@126.com

**Abstract:** Sustainable groundwater utilization is important for social and economic development. There is a need for groundwater sustainability assessment in small-scale areas lacking detailed mining data. Here, exploiting water level data series, we propose an indicator of groundwater sustainability based on the response to mining (*MGSI*) for better evaluation; it integrates groundwater data and spatio-temporal variability at a local scale. A decomposition coefficient was applied to decompose the pressure exerted by groundwater mining on the groundwater system for each monitoring well. It correlated with the groundwater response state. In Da'an City, Jilin Province, China, the appraised results revealed that the aquifer type exhibiting the greatest risk to groundwater sustainability changed from phreatic to confined during 2008–2017. The spatio-temporal distribution of different sustainability levels between and within the aquifers indicated that adjustment of the groundwater mining layout should be the focus of groundwater management in Da'an City. Additionally, the Mann–Kendall trend test and Sen's slope trend analysis effectively explained the sustainable evolution of groundwater in Da'an City and confirmed the reliability of the *MGSI* method. The proposed method highlights the effects of groundwater mining on sustainability and helps us better understand the interaction between anthropogenic activities and groundwater resources.

**Keywords:** mining; groundwater response; sustainability assessment





## 1. Introduction

Groundwater is one of the most important freshwater resources for maintaining agricultural, economic, and environmental development; it accounts for 35% of global anthropogenic water mining [1–4]. In China, 17.5% of the total water supply is through groundwater mining [5]. The domestic, industrial, and agricultural utilization rates of groundwater are 65%, 50%, and 33%, respectively, as reported by the National Groundwater Pollution and Control Plan (2011–2020) [6]. In particular, the grain-producing areas and commodity grain bases in Northeast China are dominated by groundwater irrigation, with 78% of groundwater mining used for agriculture in 2016 [7]. Consequently, the sustainable mining capacity of groundwater has become crucial to guarantee the steady growth of grain output, promote economic development, maintain the ecological environment, and ensure food security in Northeast China [8]. Accordingly, rigorous assessments of groundwater sustainability are required.

Various methods of groundwater sustainability assessment have been thoroughly investigated; however, none of them are capable of directly and accurately assessing the state of groundwater under the influence of multiple factors such as hydrometeorology, topography, and the extent of anthropogenic activities [9,10]. Thus, several indirect methods are still being applied to assess groundwater sustainability. These indirect methods are

typically based on the following: the theory of water balance and numerical simulation method [11,12], obtaining observational data by remote sensing [13], index evaluation [14], and other techniques such as machine learning [15] and the comprehensive application of multiple methods [16]. Among them, the numerical simulation method can provide effective guidance for groundwater management. However, each numerical model requires abundant datasets and the associated running time to accurately capture the complex relationships between groundwater factors. Moreover, because of the complexity of the model itself, it is difficult for managers to communicate and interact with each other and flexibly adapt to management requirements [17,18]. Remote sensing observation data, such as data obtained from the Gravity Recovery and Climate Experiment (GRACE) program, can indirectly determine the changing trend of groundwater reserves [19–22] in the evaluation of the sustainability of groundwater. However, remote-sensing-based technologies are generally more suitable for larger-scale groundwater research [23,24]. It is difficult to provide sufficient resolution for the acquisition of smaller-scale groundwater information [25]. Moreover, the inappropriate use of such data may lead to erroneous guidance for groundwater management in associated subregions [26]. In contrast, a groundwater sustainability evaluation index system based on monitoring data and statistical data can make better use of surface water data and is more suitable for local-scale assessments. Compared with traditional methods, such an index system can include multiple aspects, such as economic and aquifer variations [27]. It is considerably easier to quantify the influence of each aspect on the entire system. Many such indicators reflecting the current status and future trends of water resources have been used in resource and sustainability evaluations of groundwater and surface water [28–32]. The results have helped to improve our understanding of the spatial and temporal effects of anthropogenic activities and natural processes on water resources [33]. However, owing to the complexity of methodologies, testing, and data acquisition, such an index system is difficult for managers to use.

Long-term variations in groundwater levels reveal natural change processes and disturbances in the water budget related to anthropogenic activities (i.e., groundwater mining). Moreover, in order to describe the dynamic variations in groundwater resources, the groundwater level is more intuitive and is considerably easier to obtain than other indices applied to calculate recharge and discharge. Therefore, the primary aim of this study was to develop a method based on groundwater table data and an indicator system for groundwater sustainability assessment that is suitable for local-scale projects. Therefore, here, we propose the mining-response-based groundwater sustainability index (MGSI). In order to achieve our objective, Da'an City in Jilin Province was selected as the study area. Da'an City is an advanced grain production county (city) in China where crop irrigation is predominantly reliant on groundwater. In recent years, Da'an City has actively promoted the implementation of land consolidation and water-saving irrigation projects in order to improve the current situation of groundwater development and utilization. However, the amount of groundwater withdrawn for irrigation is still increasing (Figure A1). Therefore, a convenient and reliable method to rapidly assess the spatio-temporal distribution of local groundwater sustainability is urgently required to guide future groundwater management in Da'an City.

In this study, groundwater sustainability was quantitatively characterized by combining the pressure index imposed by anthropogenic activities with the groundwater response state. Thereafter, sustainability was classified according to the evaluation results. Subsequently, the nonparametric Mann–Kendall (MK) test [34,35] and Sen's slope trend analysis [36] were applied to analyze the significance and variation degree of the recent long-term trends of groundwater depth to verify the reliability of the index system. The index system reflects the changes in groundwater resource sustainability caused by anthropogenic activities in different locations in the region, providing an assessment method to understand the spatio-temporal evolution of local-scale groundwater sustainability and guide future mining activities.

## 2. Study Area

Da'an City is a county-level city in Baicheng City in northwestern Jilin Province, China; it experiences one of the most extreme water shortages in western Jilin [37]. The city covers an area of 4879 km$^2$ (Figure 1) and has a multi-year average precipitation of 389.2 mm and evaporation of 1702.44 mm. It is a county with scarce local surface water resources [38]. The demand for water for daily life, industrial, and agricultural activities predominantly relies on groundwater mining. There is extensive deposition of Neogene mud, sandy rocks, and loose Quaternary materials in the study area, in which several stable and superimposed aquifers have formed. Groundwater mining for industries and agriculture in the study area predominantly involves two Quaternary water aquifers, one phreatic and one confined, which were the target aquifers of this study.

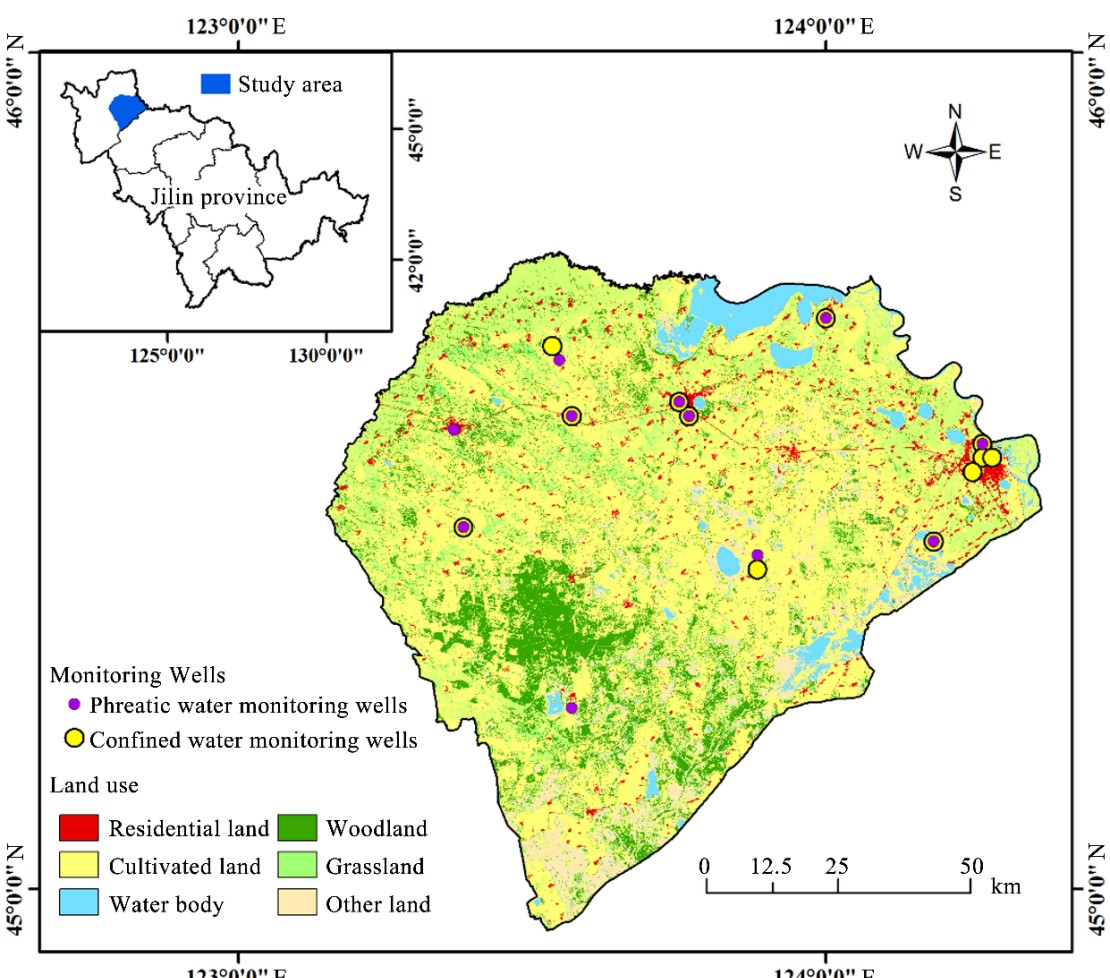

**Figure 1.** Land use and distribution of monitoring wells in the study area.

Quaternary phreatic aquifers are widely distributed in the study area. The lithology is mainly silty sand and fine sand. The thickness of the aquifer is 1–8 m. The water quality is poor and is generally alkaline. The underlying Quaternary confined aquifers are distributed throughout the area. The lithology is mainly sand and gravel, and the thickness is generally 2–30 m. The water quality is good and the mining value is large.

## 3. Data Sources

The groundwater depth data of 75 monitoring wells in the study area, collected at five-day intervals between 2000 and 2017, were obtained from the database "Groundwater Dynamic Data (Songyuan and Baicheng volume)" [39]. The monitored horizons, from top to bottom, are as follows: Quaternary phreatic water, Quaternary confined water, and

Neogene confined water. Neogene confined water was not analyzed in this study because it is considered to be a deeper confined water resource and, in principle, can only be used by special industries; therefore, data on mining and monitoring are limited for this horizon.

The data were filtered according to the completeness index (CI, the number of valid data as a percentage of the number of complete data) [40], such that the water level depth data every 5 days in each natural year for each well was greater than 75%. According to this criterion, 23 monitored wells met the requirements (Figure 1), namely, 11 Quaternary phreatic water-monitoring wells and 12 Quaternary confined water-monitoring wells. After filtering, the CI of the water level depth data for each well was 98% for 2008–2017. The groundwater mining data and precipitation data used in this study covered the period from 2000 to 2017 and were obtained from the "Baicheng City Water Resources Bulletin" [41]. The population and economic data used to calculate the degree of mining were collected from the "Jilin Statistical Yearbook" of the Jilin Provincial Bureau of Statistics [42]. The data sharing service system provided a 30 m resolution land-use map of the study area in 2020 [43].

## 4. Methodology

### 4.1. Mining-Response-Based Groundwater Sustainability Index (MGSI)

Groundwater sustainability aims to reflect the ability of groundwater to sustain long-term use [18]. Considering this concept, here, we propose the *MGSI*, which reflects both pressure of mining on groundwater sustainability and response of groundwater to mining to indicate how anthropogenic activities affect groundwater sustainability.

After determining the groundwater response state (*RES*) and mining pressure (*PRE*), the *MGSI* can be calculated using Equation (1):

$$MGSI_{ik} = RES_{ik} - PRE_{ik} \tag{1}$$

where *i* is the year of evaluation and *k* is the *k*th monitoring well.

The process of determining the *MGSI* is shown in Figure 2. In ArcGIS, the inverse distance weight method was used to interpolate groundwater sustainability in the study area, and sustainability maps of different aquifers were obtained in accordance with the different monitoring horizons. The mean *MGSI* of the entire aquifer and the *MGSI* at various locations of the aquifer can then be obtained from the *MGSI* grid graph generated by interpolation. The *RES* of each monitoring well can be calculated according to its water level depth data. The *PRE* represents the intensity of mining activities. The degree of mining in the study area is decomposed to each monitoring well through the decomposition coefficient, which indicates whether the groundwater responds sustainably to the pressure of mining activities. In this study, the calculated *MGSI* values of all aquifers were between −0.5660 and 1.938, which were quantified in ArcGIS using the Jenks Natural Breaks Classification method into the following five grades: "Low", "Relatively low", "Medium", "Relatively high", and "High" to describe the sustainability of groundwater, as shown in Table 1. Finally, the spatio-temporal variation in the groundwater sustainability level was analyzed.

**Table 1.** Classification of groundwater sustainability in the study area.

| MGSI Range | Sustainability Level of Groundwater |
|---|---|
| −0.566 to 0.148 | Low |
| 0.148–0.498 | Relatively low |
| 0.498–0.776 | Medium |
| 0.776–1.145 | Relatively high |
| 1.145–1.938 | High |

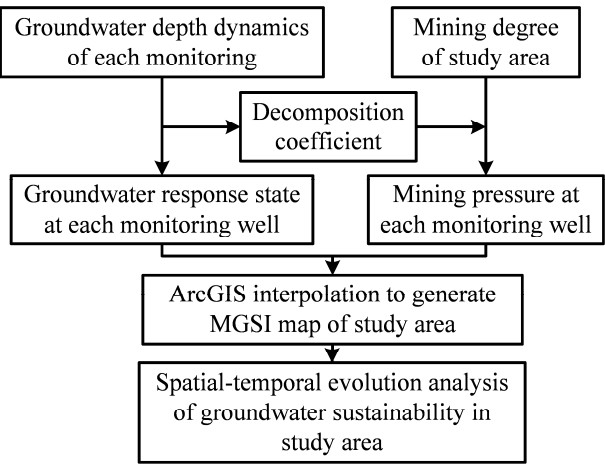

**Figure 2.** Methodological flowchart for evaluating groundwater sustainability based on artificial mining response.

*4.2. Groundwater Response State (RES)*

*RES* refers to the response of groundwater to anthropogenic activities and natural processes and is the sum of the groundwater development potential (*POT*) and groundwater reliability (*REL*), as shown in Equation (2):

$$RES_{ik} = POT_{ik} + REL_{ik} \tag{2}$$

where *POT* describes the relative distance between the groundwater depth in a certain year and the maximum historical depth. The greater the distance, the greater the development potential of the monitoring well location, as shown in Equation (3):

$$POT_{ik} = \frac{max(h_{1k}, h_{2k}, \ldots, h_{mk}) - h_{ik}}{max(h_{1k}, h_{2k}, \ldots, h_{mk}) - min(h_{1k}, h_{2k}, \ldots, h_{mk})} \tag{3}$$

According to the concept of *REL* proposed by Hashimoto et al. [44], this variable indicates the historical possibility that the system is in a satisfactory state; here, the satisfactory state refers to the rise in the water level. As shown in Equations (4) and (5), REL is the ratio of the instances of $\Delta h < 0$ and $m - 1$ in the annual water level series of each monitoring well. $\Delta h$ was calculated using Equation (5) according to the water level data, and the number of $\Delta h < 0$ indicates the "satisfactory state" in Equation (4). $m$ is the number of years in the water level series, and $m - 1$ is the number of $h$, which is "all state" in Equation (4). $h_{ik}$ is the average groundwater table depth of hole $k$ in year $i$:

$$REL_{ik} = \frac{satisfactory\ state}{all\ States} \tag{4}$$

$$\Delta h_{ik} = h_{ik} - h_{i-1,k} \tag{5}$$

*4.3. Groundwater Mining Pressure (PRE)*

The *PRE* of groundwater at each monitoring well was calculated using Equation (6):

$$PRE_{ik} = DC_{ik} \times MD_i \tag{6}$$

where $PRE_{ik}$ represents the dimensionless value of the groundwater mining pressure at the monitoring well of hole $k$ in year $i$. $DC_{ik}$ and $MD_i$ indicate decomposition coefficient (*DC*) and degree of mining (*MD*), respectively, which are explained in the following subsections.

### 4.3.1. Degree of Mining (*MD*)

The technique for order performance by similarity to ideal solution (TOPSIS)-entropy method was used to calculate the comprehensive effect of the amount and intensity of groundwater mining; the selected indexes are listed in Table 2. The entropy weight method can assign weights according to the potential information content of the data, and the TOPSIS method specifically sorts the data according to the relative closeness ($C_i$) between the evaluation object and the negative ideal solution. The annual mining degree ($MD_i$) of the study area was replaced by $C_i$, which was determined using the entropy weight TOPSIS method. Ren [45] describes these calculations in greater detail. However, to avoid a zero value, a different dimensionless method was adopted here, as shown in Equation (7):

$$y_{ij} = \frac{x_{ij}}{\sum_{i=1}^{m} x_{ij}} \tag{7}$$

where $i$ represents the 11 sample years from 2007 to 2017 and $j$ represents the seven indicators in Table 2. In addition, according to the definition of $MD_i$, the higher the frequency of groundwater mining, the greater the value of $y_{ij}$ and the greater the value of $MD_i$. The weights of each index and the calculation results of $MD_i$ are listed in Tables 2 and 3, respectively.

**Table 2.** *MD* index framework and calculation methods.

| Target | First-Level Evaluation Index | Serial Number | Secondary Evaluation Index | Data Source/ Calculation Method | Weight Relative to Target |
|---|---|---|---|---|---|
| Mining degree | Amount of groundwater mining | 1 | Total amount of mining | Baicheng City Water Resources Bulletin | 0.084 |
| | | 2 | Amount of mining for irrigation | Baicheng City Water Resources Bulletin | 0.109 |
| | | 3 | Amount of mining for industry | Baicheng City Water Resources Bulletin | 0.122 |
| | Intensity of groundwater mining | 4 | Groundwater consumption per 10,000 Yuan of GDP | Total amount of mining (10,000 m$^3$)/GDP (10,000 Yuan) | 0.207 |
| | | 5 | Groundwater consumption per capita | Total amount of mining (10,000 m$^3$)/total population (10,000) | 0.099 |
| | | 6 | Intensity of mining for agriculture | Amount of mining for irrigation (10,000 m$^3$)/total number of agricultural wells | 0.228 |
| | | 7 | Groundwater consumption per 10,000 Yuan of industrial production value | Amount of mining for industry (10,000 m$^3$)/industrial production value (10,000 Yuan) | 0.151 |

**Table 3.** *MD* calculation results.

| $i$ | $MD_i$ | $i$ | $MD_i$ | $i$ | $MD_i$ |
|---|---|---|---|---|---|
| 2007 | 0.126 | 2008 | 0.063 | 2009 | 0.626 |
| 2010 | 0.789 | 2011 | 0.390 | 2012 | 0.392 |
| 2013 | 0.140 | 2014 | 0.303 | 2015 | 0.487 |
| 2016 | 0.971 | 2017 | 0.966 | | |

### 4.3.2. Decomposition Coefficient (*DC*)

As the *MD* can only describe the state of groundwater mining in the entire region, decomposition of *MD* is required to further reflect the *PRE* of the local area and different aquifers. Water level depth dynamics help us understand this groundwater balance as

the difference between the water level depth variation amplitude of different monitoring wells reflecting the magnitude of the production pressure in different locations. Therefore, the *DC*, calculated using Equation (8), was proposed to decompose the *MD* into different spatial locations and closely correlate groundwater mining with the groundwater response. The symbols and subscripts in Equation (8) have the same meaning as those in Equation (5):

$$DC_{ik} = \frac{\Delta h_{ik} - min\left(\Delta h_{i1}, \Delta h_{i2}, \ldots, \Delta h_{iq}\right)}{max\left(\Delta h_{i1}, \Delta h_{i2}, \ldots, \Delta h_{iq}\right) - min\left(\Delta h_{i1}, \Delta h_{i2}, \ldots, \Delta h_{iq}\right)} \tag{8}$$

*4.4. Trend Test*

The MK trend test method and Sen's slope method were used to detect the trends of time series data and the intensity of trend changes, respectively. These nonparametric methods are widely used in the field of hydrometeorology because they do not require testing of data distribution and the results are considered reliable [46–51]. Here, the two methods were combined to test the annual variation trend of the mean *MGSI* in the study area and the annual average groundwater depth trend of each monitoring well from 2007 to 2017. Details of these methods are not provided here; however, full descriptions can be found in the aforementioned references.

## 5. Results and Discussion

*5.1. Spatio-Temporal Variation in Groundwater Sustainability*

The mean *MGSI* of confined water exhibited the same change trend as the amount of mining (Figure 3), indicating that the sustainability of confined water is substantially affected by groundwater mining. Prior to 2015, the *MGSI* of confined water was higher than that of phreatic water; however, later, it became lower than that of phreatic water, although the difference was relatively marginal. This phenomenon was aggravated in 2016–2017, indicating that the rapid increase in mining over the previous years placed a burden on confined water, which is not conducive to groundwater conservation. Therefore, Da'an City should formulate a more reasonable mining plan to ensure the sustainable use of groundwater resources.

In comparison, the phreatic water trend more closely followed that of precipitation, although it was also slightly affected by the amount of mining (Figure 3). For example, during 2012–2014, a decrease in precipitation led to a decrease in the recharge received by phreatic water. However, owing to the simultaneous decrease in the amount of mining during this period, the decreased precipitation did not ultimately lead to a downward trend in the mean *MGSI* of phreatic water. The same situation occurred during 2014–2016, wherein precipitation increased the recharge of phreatic water, but a significant increase in mining ultimately caused a decrease in the mean *MGSI* of phreatic water.

The actual state of groundwater in Da'an City is consistent with the results of this study, that is, poor water quality and low mining are characteristics of phreatic water in the study area and precipitation is the main factor affecting the phreatic water. Although the changes in precipitation affect the leakage recharge of phreatic water to the confined aquifer, confined water is still the main source of water in Da'an City, and it is largely influenced by anthropogenic factors.

The highest mean *MGSI* values for the confined aquifer (1.326) and phreatic aquifer (0.999) occurred in 2008 and 2014, respectively. The lowest mean *MGSI* values for both aquifers occurred in 2010 (0.230 and 0.283 for phreatic and confined water, respectively). The calculation results of the MK trend test and Sen's slope analysis are shown in Table 4. Notably, the average *MGSI* of the phreatic aquifer increased at a rate of 0.01 per year, whereas that of the confined aquifer decreased at a rate of 0.04 per year, which indicates that the risk of groundwater becoming nonsustainable within the study area shifted from the phreatic aquifer to the confined aquifer with time. However, this trend was not significant at the 95% confidence level.

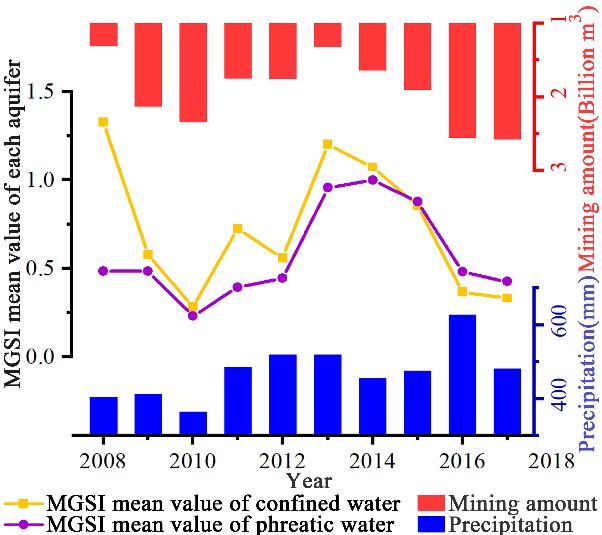

**Figure 3.** Annual variation trends of the mean *MGSI* for the phreatic and confined aquifers according to the amount of mining and precipitation.

**Table 4.** Trends of mean *MGSI* for aquifers in the study area.

| Aquifer | Mean *MGSI* | Significance (5% Significance Level) | Trend | Sen's Slope (/a) |
|---|---|---|---|---|
| Phreatic | 0.230–0.999 | Insignificant | Increasing | 0.01 |
| Confined | 0.283–1.326 | Insignificant | Decreasing | −0.04 |

This finding is supported by the observation that the zone with the lowest sustainability was the largest in the phreatic aquifer in 2010, representing 45.2% of the study area, but also was the largest in the confined aquifer in 2017, representing 32.5% of the study area (Figure 4). In 2008, the sustainability levels of the two aquifers were highly in contrast. This phenomenon can be explained by the fact that the annual variation in groundwater depth was used to calculate the MGSI; that is, the groundwater depth in 2007 was used to calculate the *MGSI* in 2008. To confirm the validity of the calculation results for 2008, the variation in water level was calculated for 23 monitoring wells from 2007 to 2008. During this period, the groundwater depth of phreatic monitoring wells increased by 0.52 m per year, whereas that of the confined aquifer decreased by 0.03 m per year. This explains the difference in the sustainability levels of the two aquifers in 2008.

During 2009–2012, the sustainability level of the phreatic aquifer was predominantly "Relatively low", whereas that of the confined aquifer was predominantly "Medium". During this period, the depth of the confined aquifer was relatively stable and the overall sustainability was better than that of the phreatic aquifer. Similarly, during 2013–2015, both aquifers showed larger areas of "High" and "Relatively high" sustainability. However, subzones of "High" and "Relatively high" sustainability in the confined and phreatic aquifers gradually decreased after 2013 and 2014, respectively. After 2015, "Low" sustainability subzones gradually expanded, particularly in the confined aquifer.

To understand the changes in the spatio-temporal distribution of the sustainability level of each aquifer, representative years were selected (2008, 2010, 2014, and 2017). Figure 5 shows the spatial changes in the sustainability level subzones of each aquifer throughout the study period (2008–2017). In 2008, "Relatively low" sustainability subzones accounted for a relatively large proportion of the phreatic aquifer (Figure 5a), and they were distributed in the eastern and central regions (Subregion a1 in Figure 5a). This was almost the worst state of phreatic sustainability in 2008, except for a localized "Low" sustainability subzone in the northeast of the study area (Subregion a2 in Figure 5a). By 2010, the

"Medium" sustainability subzone in the east (Subregion a1) had expanded into the middle (Subregion b1 in Figure 5b) of the study area, with some "Relatively high" sustainability subzones appearing in this region. In 2007, in order to secure food and water supplies, the Jilin Provincial Government began to implement the use of water from the Nenjiang River (in the eastern and central parts of the study area) to irrigate the Da'an Irrigation Area. This resolution used part of the confined aquifer area for irrigation and transformed dry and saline land into paddy fields [52]. The expansion of paddy fields would have increased the amount of infiltration into the phreatic aquifer, thereby raising the water level in the area. Dry conditions in 2010 then encouraged evaporation, and high groundwater consumption increased mining, leading to "Low" sustainability levels over a relatively large area of the phreatic aquifer in this year. High precipitation and low mining around 2014 then effectively replenished the groundwater resources, which had an equally large effect on both phreatic and confined aquifers (Figure 5c,g). Continuously increasing mining resulted in the "Relatively low" sustainability subzone again occupying most of the phreatic aquifer (Figure 5d).

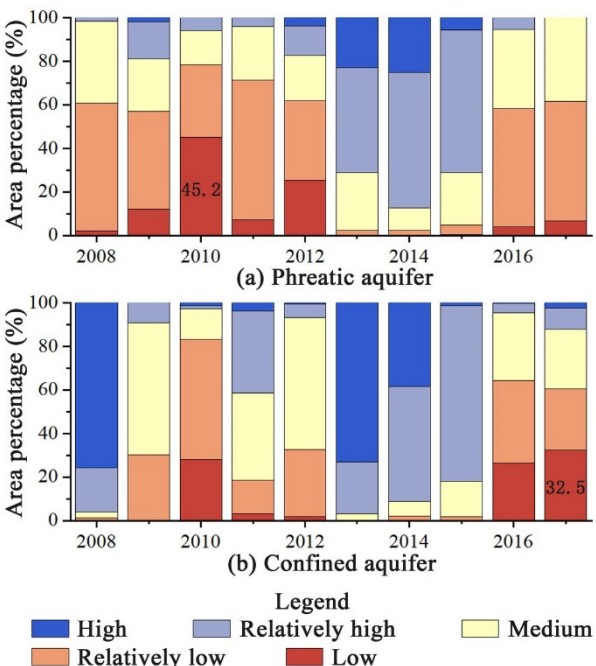

**Figure 4.** Percentage (by area) of different groundwater sustainability subzones in the study area.

Considering that confined water level recovery in 2008 was relative to the previous year, the sustainability level for the confined aquifer in that year was "High" (Figure 5e). By 2010, under the influence of a more arid climate, almost the entire area exhibited "Relatively low" sustainability, with a large area of "Low" sustainability appearing in the center of the study area (Subregion f1 in Figure 5f). This is because the Da'an Irrigation Area is located in the central region (Subregion f1), which contained a large number of paddy fields that were irrigated by the Nenjiang River in normal years and supplemented by confined water in dry years. Low precipitation in 2010 and crop growth demands led to highly concentrated mining, thereby contributing to the development of a "Low" sustainability subzone in the central region (Subregion f1). To verify the above explanation, the actual amount of groundwater mining was investigated over the entire study area used for agricultural irrigation. The data showed that phreatic water was not used for agricultural irrigation. The amount of confined water extracted for agricultural irrigation was 181.85 million $m^3$ in 2009, 146.37 million $m^3$ in 2011, when rainfall was relatively abundant, and 200 million $m^3$ in 2010, when rainfall was relatively infrequent. The increased mining of confined water in 2010 effectively explain the "Low" sustainability subzone in the central region (Subregion

f1). With this reduction in confined water sustainability, "Low" sustainability subzones became concentrated in the west of the study area by 2017 (Subregion h1 in Figure 5h).

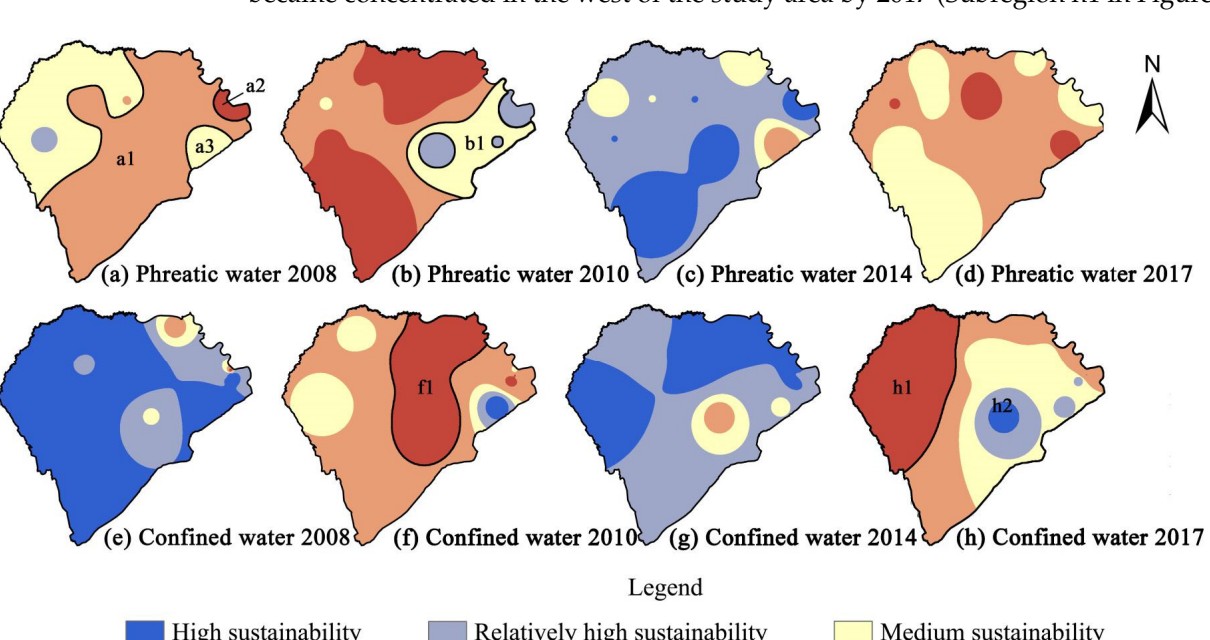

**Figure 5.** Distribution of sustainability level subzones.

Currently, industrial, agricultural, and domestic water use in Da'an City mainly depend on groundwater, and confined water accounts for a large proportion. Therefore, strengthening the protection of groundwater resources is important for economic development, food security, and water security. In the future, Da'an City should pay close attention to groundwater mining in the western region and improve water use efficiency throughout the region to avoid uncontrolled mining.

*5.2. Change Trends of Groundwater Depth*

According to the calculation results of the groundwater depth change trend, only a small portion of phreatic monitoring wells (27.3%, three wells) exhibited a decreasing trend during 2007–2017. Phreatic water depth mainly increased in the north, at a typical rate of 0.036–0.083 m/a (Figure 6). The difference in groundwater depth between the north and south verified the results shown in Figure 5, i.e., that the phreatic sustainability level was typically higher in the south than in the north. According to the distribution of Sen's slope values (Figure 6), the trend of rising groundwater depths in southern (−0.009 to 0.036 m/a) and northeastern marginal areas (−0.077 to 0.009 m/a) was consistent with the long-term trends of the distribution of sustainability level subzones for phreatic water. The results for the northeastern marginal area are the most intuitive, that is, the area experienced "Low", then "Relatively high", then "High" sustainability levels during the study period (the subregion corresponding to a2 in Figure 5a–c). By 2017, although the sustainability level of the area had decreased, it was still higher than that of most other areas (Figure 5d). Moreover, Sen's slope value (0.083–0.308 m/a) at Well 9 in Figure 6 verifies the change from "Relatively high" sustainability to "Relatively low" and then "Low" sustainability (Figure 5b–d). In addition, for phreatic water, the number of monitoring wells with significant trends was very small (18.2%). Only two phreatic water-monitoring wells exhibited a significant increase in depth during the study period, namely, Wells 8 and 9 (Figure 6). Of these, Well 9 (southeast of the study area) exhibited the largest Sen's slope value of the entire area (Figure 6), with a significant increase in depth of 0.31 m/a, whereas that for Well 8 (northwest) was 0.15 m/a. These significant increasing trends of groundwater depth indicate that the phreatic water in these subzones has been over-

consumed in recent years. Therefore, long-term monitoring in these subzones is necessary to maintain the sustainable utilization of groundwater.

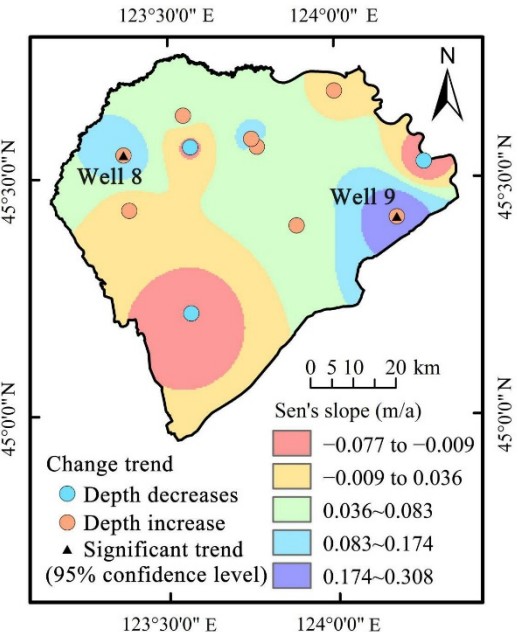

**Figure 6.** Distribution of Sen's slope values for phreatic water depth and water depth trends in the monitoring wells.

Figure 7 shows the confined water trend analysis and spatial distribution of the Sen's slope calculation results. During the analysis period, six of the twelve confined water-monitoring wells showed an increasing trend, whereas the other six showed a decreasing trend. In contrast to phreatic water, the confined water depth trend differed in the east-west direction, that is, it increased in the west of the study area but generally decreased in the east. The area of increasing groundwater depth (Sen's slope value of 0.015–0.328 m/a in Figure 7) agrees with the reduction in the sustainability level of confined water observed in the west of Da'an City in Figure 5, where the dominant sustainability level in this region changed from "High" to "Low" (subregion corresponding to h1 in Figure 5e,f,h). However, the sustainability level of confined water in the east of the study area (subregion corresponding to h2 in Figure 5e–h) was not consistent with the trend of Sen's slope values (Sen's slope value of −0.316−−0.030 m/a in Figure 7) for the groundwater depth in the monitoring wells. Specifically, the groundwater depth in the monitoring wells in the area characterized by the Sen's slope values of 0.034–0.328 showed a consistent increasing trend, whereas that in the area with the Sen's slope values of −0.316 to 0.034 showed a different trend. Therefore, the spatio-temporal variation in the confined water sustainability level is more complex in the east of the study area than in the west. Regarding the variation in confined water level depths, 16.7% of the monitoring wells (two wells) showed a significant increasing trend. In contrast, 8.3% (one well) showed a significant decreasing trend. Figure 7 shows three confined water-monitoring wells with significant trends in the study area, where Well 5 exhibited the fastest decrease in water depth over the study area (0.32 m/a) and Well 18 exhibited the largest increase in water depth over the study area (0.33 m/a). The significant decrease in water depth in Well 5 indicates that the confined aquifer in this area has the potential for further mining.

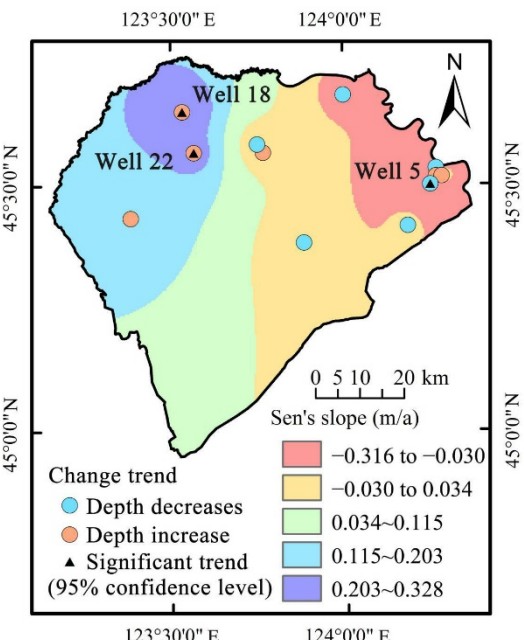

**Figure 7.** Distribution of Sen's slope values for confined water depth and water depth trends in the monitoring wells.

## 6. Conclusions

We demonstrated a method developed to assess the sustainability of local-scale groundwater based on the degree of groundwater mining and the subsequent changes in groundwater level, called the mining-response-based groundwater sustainability index (MGSI). This index can be used by water management departments to guide groundwater development and utilization planning. The selection of indicators and evaluation methods is a key aspect of sustainability evaluations. We introduced the following new indicators to construct the evaluation system: (1) the decomposition coefficient (DC) was proposed in order to decompose the degree of mining over the entire study area into mining pressure at each point; (2) an indicator of the development potential of groundwater (POT) was used to describe the relative distance between the current water depth and the maximum historical depth; and (3) an indicator was developed to reflect the reliability of groundwater in terms of the potential groundwater level rise (REL). Long-term variation trends of groundwater depth verified the reliability of the *MGSI* method. The proposed groundwater sustainability index based on groundwater's response to mining can be used to monitor groundwater sustainability and analyze local spatio-temporal variations in groundwater sustainability. Such an analysis can help identify areas of potential groundwater sustainability and areas requiring protection from groundwater mining, as well as to implement effective groundwater resource management.

We demonstrated the potential of the evaluation method by taking Da'an City as an example and made the following conclusions:

(1) The mean *MGSI* of confined water is more significantly affected by groundwater mining than that of phreatic water. During 2013–2017, with the increase in mining, the mean *MGSI* of confined water dropped sharply, and the mean *MGSI* of phreatic water also showed a similar trend. In the future, water resource management should consider replacing groundwater with surface water or other water sources to reduce groundwater exploitation.

(2) The mean *MGSI* of the phreatic aquifer increases at a rate of 0.01 per year, whereas that of the confined aquifer decreases at a rate of 0.04 per year. Therefore, the mining of confined water in Da'an City should be carried out more cautiously in the future.

(3) The sustainability level evaluation using the *MGSI* shows that the continuous increase in mining up to 2017 subsequently led to the "Relatively low" sustainable subzone

occupying most of the submerged aquifer, and the "Low" sustainable subzone of the confined aquifer was concentrated in the west of Da'an City. Reducing mining and the utilization of groundwater is still a challenge that should be overcome in Da'an City, especially in the west of Da'an City. Taking into account the local conditions, developing water-saving agriculture, or replacing paddy fields with dry fields should be considered.

(4) The groundwater depths of Well 8 and Well 9 have increased significantly, indicating that the phreatic water in these areas has been overconsumed during recent years, and management measures should be implemented in the future while continuing to pay attention to the groundwater levels in these areas. The groundwater depth of Well 5 decreased significantly, indicating that the confined water in this area has the potential for further mining.

**Author Contributions:** Z.F. proposed the indicator of *MGSI* and drafted the manuscript. X.D. appraised the groundwater sustainability. H.G. drew the figures and reviewed the manuscript. All authors have read and agreed to the published version of the manuscript.

**Funding:** This research was funded by the National Key Research and Development Program of China under Contract No. 2020YFC1808300.

**Institutional Review Board Statement:** This study did not require ethical approval; all data are available in the public domain.

**Data Availability Statement:** Authors understand that journal encouraged data sharing and data citation.

**Conflicts of Interest:** The authors declare no conflict of interest.

## Appendix A

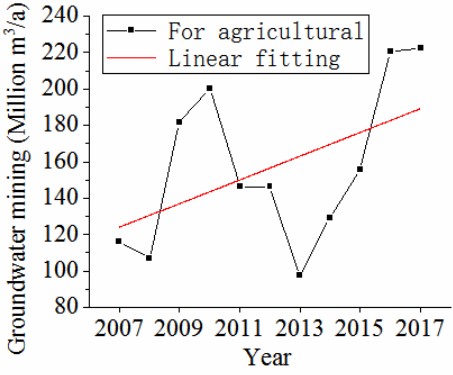

**Figure A1.** Agricultural exploitation from 2007 to 2017 and linear fitting.

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
