# Peer review of "Local-Scale Groundwater Sustainability Assessment Based on the Response to Groundwater Mining (MGSI): A Case Study of Da’an City, Jilin Province, China"

_sustainability, doi:10.3390/su14095618_

Round 1

Reviewer 1 Report

  1. Authors should explain what does “AGSI” mean in line 151.
  2. There is no description of the sampling and analysis method in “RELik (line 183).
  3. Line 19: Title: It is not clear, "the spatio-temporal distribution of different sustainability levels between…”. How can analysis be performed in the future groundwater management in Da'an City?
  4. This conclusion is not enough supported by evidences for implementing effective groundwater resource management. Authors should mention more.

Author Response

1: Authors should explain what does “AGSI” mean in line 151.

Reply: We have changed "AGSI" to "MGSI" in Equation 1. We apologize for the oversight.

2: There is no description of the sampling and analysis method in “RELik (line 183).

Reply: We have added a detailed description of how Equation (4) is obtained (indicated in red font in the manuscript).

3: Line 19: Title: It is not clear, "the spatio-temporal distribution of different sustainability levels between…”. How can analysis be performed in the future groundwater management in Da'an City?

Reply: We have added the corresponding description in the conclusion per your comment.

4: This conclusion is not enough supported by evidences for implementing effective groundwater resource management. Authors should mention more.

Reply: We thank you for the comment. We have revised the relevant portion accordingly.

Reviewer 2 Report

The manuscript present the development of new index/indicator for the assessment of groundwater sustainability. The indicator was apply to a case study in China based on data from 23 wells. The manuscript is well written, easy to read with appropriate amount and structure of literature. There is used mainly current literature published after 2015 year.

Weaknesses of the manuscript:

  • it is not clear what the authors mean by water mining, water withdrawal, water extraction and water exploitation - it would be fine to add meaning of terms by authors to the text
  • the key words are missing
  • line 194 - should be reference for Table 2
  • Table 1 - it is not clear how the range was created
  • there is some misunderstanding between sentece in lines 162-165 ("In this study, the calculated MGSI values of all aquifers were between -0.5660 and 0.148, which were quantified into the following five grades: “Low,” “Relatively low,” “Medium,” “Relatively high,” and “High” to describe the sustainability of groundwater, as shown in Table 1"), information presented in Table 1 and presented results

Author Response

1: It is not clear what the authors mean by water mining, water withdrawal, water extraction and water exploitation - it would be fine to add meaning of terms by authors to the text.

Reply: We reviewed the use of "water mining", "water withdrawal", "water extraction" and "water exploitation" in the manuscript, and decided to use "water mining" uniformly.

2: The key words are missing.

Reply: We thank you for poiting this out. We have added the keywords—"Extraction; Groundwater response; Sustainability assessment."

3: Line 194 - should be reference for Table 2.

Reply: We thank you for the comment. We have made the necessary revision in the manuscript.

4: Table 1 - it is not clear how the range was created.

Reply: Jenks Natural Breaks Classification method was used in ArcGIS for classification, and the description has been added in the manuscript.

5: There is some misunderstanding between sentece in lines 162-165 ("In this study, the calculated MGSI values of all aquifers were between -0.5660 and 0.148, which were quantified into the following five grades: “Low,” “Relatively low,” “Medium,” “Relatively high,” and “High” to describe the sustainability of groundwater, as shown in Table 1"), information presented in Table 1 and presented results

Reply: As mentioned in our response to Point 4, Jenks Natural Breaks Classification method was used in ArcGIS for classification, and the description has been added in the manuscript.

Reviewer 3 Report

I congratulate the Authors for their interesting paper, in which methods, interpretation and results are clearly presented and summarized in the conclusions. All data are consistent with each other and allow to deepen assesment of local scaled sroundwater sustainability.

In the reviewed report I suggested some small changes. I hope that the comments will be useful to the authors to further enhance the quality of their paper.

There are many typographic and spelling errors throughout the manuscript. Please, revise the paper carefully.

Small description of elementary aquifers of the multi-layer system is needed.

In the introduction section; the novelty of the work must be established.

Equation 1: I gues AGSI should MGSI? If not please clarify?

Figure 3: Please rectify the legend. Red rectangle represent mining amout not precipitation. 

The conclusion must be rewritten with the main results.

Author Response

1: There are many typographic and spelling errors throughout the manuscript. Please, revise the paper carefully.

Reply: The revised manuscript was edited by Editage.

2: Small description of elementary aquifers of the multi-layer system is needed.

Reply: Short descriptions of the two target aquifers have been added per your suggestion.

3: In the introduction section; the novelty of the work must be established.

Reply: The novelty of this work lies in the full use of water level monitoring data to enable sustainable groundwater evaluations in small-scale areas that lack detailed extraction information. This has been explained in the introduction section.

4: Equation 1: I gues AGSI should MGSI? If not please clarify?

Reply: Yes, we have corrected the mistake here. We apologize for the oversight.

5: Figure 3: Please rectify the legend. Red rectangle represent mining amout not precipitation.

Reply: We apologize for the oversight. We have fixed the mistake.

6: The conclusion must be rewritten with the main results.

Reply: Per your suggestion, we have added the conclusions based on the major results as follows:

We demonstrated the potential of the evaluation method by taking Da'an City as an example and made the following conclusions:

(1) The mean MGSI of confined water is more significantly affected by groundwater mining than that of phreatic water. During 2013–2017, with the increase in mining, the mean MGSI of confined water dropped sharply, and the mean MGSI of phreatic water also showed a similar trend. In the future, water resource management should consider re-placing groundwater with surface water or other water sources to reduce groundwater ex-ploitation.

(2) The mean MGSI of phreatic aquifer increases at a rate of 0.01 per year, whereas that of confined aquifer decreases at a rate of 0.04 per year. Therefore, the mining of con-fined water in Da'an City should be carried out more cautiously in the future.

(3) The sustainability level evaluation using the MGSI show that the continuous in-crease in mining up to 2017 subsequently led to the “Relatively low” sustainable subzone occupying most of the submerged aquifer, and the “Low” sustainable subzone of confined aquifer was concentrated in the west of Da'an City. Reducing mining and utilization of groundwater is still a challenge that should be overcome in Da'an City, especially in the west of Da'an City. Taking into account the local conditions, developing water-saving ag-riculture or replacing paddy fields with dry fields should be considered.

(4) The groundwater depth of Well 8 and Well 9 has increased significantly, indicat-ing that the phreatic water in these areas has been overconsumed during recent years, and management measures should be implemented in the future while continuing to pay at-tention to the groundwater levels in these areas. The groundwater depth of Well 5 de-creased significantly, indicating that the confined water in this area has the potential for further mining.

Round 2

Reviewer 1 Report

Authors have corrected these revisions according to my opinions.

Author Response

Thank you for your recognition.

Reviewer 2 Report

I appreciate work done by authors to improve the manuscript. However, I still have a problem with information presented in lines 340-355: "In this study, the calculated MGSI values of all aquifers were between -0.5660 and 0.148, which were quantified in ArcGIS using the Jenks Natural Breaks Classification method into the following five grades: “Low,” “Relatively low,” “Medium,” “Relatively high,” and “High” to describe the sustainability of groundwater, as shown in Table 1.” and Table 1, where the maximum value is 1.938. So, the range is -0.5660 and 0.148 or -0.5660 and 1.938? According to Table 1, only “Low” is in the range -0.5660 and 0.148, but not all calculated values. Please, revised it once again and write is clearer.
